# Waste Separation in Cafeterias: A Study among University Students in the Netherlands

**DOI:** 10.3390/ijerph16010093

**Published:** 2018-12-31

**Authors:** Ágústa D. Árnadóttir, Gerjo Kok, Suzanne van Gils, Gill A. ten Hoor

**Affiliations:** Department of Work & Social Psychology, Maastricht University, PO BOX 616, 6200MD Maastricht, The Netherlands; a.arnadottir@student.maastrichtuniversity.nl (Á.D.Á.); g.kok@maastrichtuniversity.nl (G.K.); suzanne.vangils@maastrichtuniversity.nl (S.v.G.)

**Keywords:** recycling, waste separation behavior, determinants of behavior, intervention, university students

## Abstract

Recycling waste is important to reduce the production of greenhouse gasses. The aim of this project was to understand determinants of cafeteria waste separation behavior among university students. First, the determinants of waste separation behavior among university students (*n* = 121) were explored using an online questionnaire. In study 2 (pre-/post-test design), the effect of a small intervention (based on study 1) on actual waste sorting behavior was observed. Finally, a semi-qualitative study in 59 students was conducted as process evaluation of the intervention. The following results were revealed: (1) Students have limited knowledge about waste separation, have a high intention to separate waste, are positive about waste separation in general, and believe that they can separate waste correctly. (2) Just over half of the waste is correctly recycled. An intervention with extra information had no significant effect on improving recycling behavior. (3) Students evaluated the intervention positively. Some students suggested that more information should be available where the actual decision making takes place. Ultimately, this paper concludes that although students have a positive attitude and are willing to behave pro-environmentally, there is a gap between intention and actual behavior. These results may also apply to other organizations and members of those organizations. New interventions are needed to trigger students to make correct waste separation decisions where the actual decision making takes place.

## 1. Introduction

In 2016, nearly 9 million tons of waste was produced in the Netherlands. About half of that waste ended up in incineration and recovery, with its accompanying CO_2_ emissions [1]. To decrease the CO_2_ emissions, the European Union (EU) has set a goal for the countries of the EU and the European Economic Area (EEA) to recycle at least 50% of municipal waste by the year 2020, 60% by the year 2025, and 65% by the year 2030. Although the Netherlands succeeded in the first goal (52% of the Dutch municipal waste was recycled or reused), there are still concerns: recycling in the Netherlands increased by 5% from 2004–2015 [2], making it unlikely that the 2025–2030 goals will be reached when this trend continues. Different initiatives have started to improve waste separation in the Netherlands. For example, the city of Maastricht is doing its part by having the ambition to completely eliminate unsorted household waste by 2030. In line with the city’s goals, Maastricht University (UM) has published its vision to become a sustainable university by the year 2030. This includes making important contributions towards a sustainable future through education and research, being aware of the University’s social, economic, and environmental impacts, and taking sustainability into account in all decision making [3]. One of the factors mentioned to reach that goal is to reduce waste production and to increase recycling. In 2016, students and staff produced 20 kg of waste each on the UM campus. The University has set the goal of cutting down on general waste from 55% to 45% of the total waste by 2021 [4]. In order to reach that goal, the University introduced new waste sorting stations on campus in 2017. These stations include bins for sorting waste into paper, plastic, and general waste [5].

In 2017, the UM Green Office [6] conducted a short survey to explore how well the new waste sorting system was communicated to the students, and to examine the current knowledge about correctly sorting their waste (*n* = 222). The main conclusion was that communication on the recycling guidelines should be improved. On average, 6–6.5 out of 10 products were sorted correctly. However, there were limitations to this initial study: first, there were indications that students ‘cheated’ by looking up answers to increase their chances of winning a prize, indicating that the results of waste sorting knowledge might not have reflected actual knowledge. Second, actual sorting behavior and behavioral determinants were not measured. Lastly, although it was likely that the majority of the participants were students, also other people (scientific staff, cafeteria personnel, visitors, etc.) could have participated, influencing the results. To overcome these problems, and to increase our understanding of waste separation among university students, we conducted three studies to (1) identify the relevant determinants, (2) observe actual waste separation behavior, and (3) pilot test an intervention targeting the identified determinants. In the first study, behavioral determinants, intentions, possible barriers, and current knowledge about cafeteria waste separation in university students were examined by applying the Reasoned Action Approach (RAA) [7]. The RAA and the closely related Theory of Planned Behavior (TPB) model has often been used as the basis for measuring determinants of recycling (and other pro-environmental behavior) in various settings, although not in the specific setting of the current studies (e.g., Greek citizens’ recycling behavior [8], recycling behavior among Australian students [9], pro-environmental behavior among high school students in Luxembourg [10], separation of food waste among university staff in Malaysia [11], household food waste reduction among UK residents [12], and waste separation behavior among citizens of Guangzhou, China [13]). In the second study, a small-scale intervention was conducted and actual waste separation behavior in the cafeteria was observed before and after the intervention to determine its effect. Lastly, we conducted a process evaluation to evaluate how the intervention was processed and to what extend the intervention was accepted.

## 2. Study 1: Determinants of Waste Separation 

### 2.1. Background/Methods—Study 1

In the first study, behavioral determinants, intentions, possible barriers, and current knowledge in relation to waste separation in university students were examined, based on the RAA [7].

Participants in the study consisted of 121 students from a Dutch university. Participants, approached in person at university during one weekday between noon and 2:00 p.m., and through various social media outlets, were asked to fill out a short online questionnaire. All subjects gave their informed consent for inclusion before they participated in the study. The study was conducted in accordance with the Declaration of Helsinki, and the protocol was approved by the Ethics Review Committee Psychology and Neuroscience (ERCPN), Maastricht University, the Netherlands (ERCPN 188_11_02_2018_S1).

#### The Questionnaire

After receiving the relevant information at the start of the survey, participants confirmed their informed consent by pressing the appropriate button. Next, participants filled out an online questionnaire that consisted of 29 questions, divided into four parts. First, demographic questions were asked (gender, age, whether the participant was a student/staff/other). Subsequently, to test the students’ waste sorting knowledge, they were asked to virtually sort 15 pictures of products (as sold in the UM cafeteria) into either a paper, plastic, or general waste bin (see Appendix A). One item (a sandwich bag with a sandwich in it) was deleted, as participants indicated that this picture was confusing. In the third part of the questionnaire, participants were first shown the correct answers from the waste sorting (based on the University guidelines), before answering questions about behavioral determinants, barriers, and waste separation intention. The questions were based on the Reasoned Action Approach (Appendix A in [7]). We used the short version of the standard questions for the relevant concepts from the theory. All these questions were answered on a 7-point scale with the advised end points (Appendix A in [7]). Scores that measured the same construct were averaged into one scale where internal consistency was sufficient (a > 0.60). Scores were recoded such that a higher score reflected a higher value on the variable. All questions can be found in Table 1. Lastly, an open-ended question was used to ask for suggestions on how participants would like to receive information about changes in the waste sorting.

### 2.2. Results—Study 1

From the 121 participants, 15 participants were excluded from the analysis: 9 for submitting partial responses (missing answers to >1 questions), and 6 for either answering that they were not students at the university or failing to provide that information. Therefore, answers from 106 participants were used for further analyses. In total, 77.4% of the participants were female, with a mean age of 23 years, and age ranged from 18–50 years old.

Knowledge about waste separation—On average, participants answered 67.7% of the questions correctly (on average 9.5/12 items). The findings show that there is a large range of knowledge with regard to how to sort items, which is contingent on the type of product. While the knowledge for most plastics and food waste was (close to) 100%, for other items, such as coffee cups, sugar sticks, and candy wrappers, the knowledge demonstrated was suboptimal, and almost all participants thought, erroneously, that napkins could be separated as paper (see Table 2).

*Behavioral determinants related to waste separation*—On average, the university students had a positive intention to separate waste (*M* = 6.30/7; *SD* = 0.76), were positive about waste separation in general (*M* = 5.78/7; *SD =* 0.99), experienced a positive subjective norm (*M* = 5.13/7; *SD* = 1.65), and believed that they could separate waste correctly (*M* = 5.51/7; *SD* = 1.15), as depicted in Table 3. All these determinants are positively correlated to intention, which is in line with other comparable studies [8,9,10,11,12,13]. Contrary, students were less positive about other students separating their waste correctly (*M* = 3.78/7; *SD* = 1.38), but this was not correlated to their own intention to separate their waste. Together, the behavioral determinants predicted 62% of a student’s waste separation intentions (see Table 3), which is again in line with other comparable studies [8,9,10,11,12,13]. Adding past behavior to the regression improved the prediction of intention but not by much (66%; R-square change was significant, *p* = 0.001).

Barriers and suggestions—the two open-ended questions asking participants about potential barriers regarding waste separation led to 110 answers. These were separated into reasons for not separating waste correctly, and suggestions: 68 answers were related to unclear waste separation guidelines, confusion about good waste separation, or little knowledge (one out of those 68 disagreed with the University guidelines); 15 students stated that that there were not enough (different) bins (*n* = 13) to separate all their waste, or that the available containers were dirty (*n* = 2). A total of 18 students did not separate waste because of time pressure (*n* = 9) or laziness (*n* = 9); 2 students explained that they were not willing to tear apart packaging that consisted of both a paper and plastic part; 1 student explained “I’m not sure it makes a significant difference if I sort 90% of my trash correctly or 100% and can’t be bothered to invest the effort for the extra 10%.” A few students suggested to add tags to the products (*n* = 1), or informational pictures on the bins (*n* = 4).

### 2.3. Conclusions—Study 1

In Study 1 the main focus was to investigate the effects of the determinants of behavior as specified by the RAA model [7] on intention to sort the waste. The results from this first study showed that all three determinants significantly relate to intention in the regression model; self-efficacy was observed to have the largest effect. From the qualitative analysis, it appears that although intention to recycle was moderate to high, knowledge about the correct way to sort the waste was insufficient for certain categories of items. Intention is often a good predictor of behavior, but not always, especially not when people do not have enough knowledge or skills to perform the behavior correctly [7,14]. Moreover, the absence of an influence from descriptive norms suggests that information about others separating waste correctly may also increase actual behavior. To address the first of these determinants, we designed a behavioral intervention in Study 2, with the aim to increase knowledge and skills of how to sort waste correctly. Influencing descriptive norms is difficult with behavior that can be observed by the target population, and social norms should only be mobilized when they are clearly supportive of the behavior [15].

## 3. Study 2: Waste Separation Behavior and the Effects of an Intervention to Improve Waste Separation among University Students

### 3.1. Background/Methods—Study 2 

The results of study 1 suggest that the ‘how to do’ knowledge about waste separation was suboptimal, while students intentions were high. Improving knowledge seemed to be the most logical determinant to improve first. The intervention that was developed was based on the idea of nudging [16], as applied in earlier studies on interventions to improve recycling behavior [17,18,19].

Therefore, for one week, actual waste sorting behavior during lunch hours (11.00–2.00 p.m.) was observed at the Maastricht University health campus cafeteria. During the observation hours, one waste separation station was observed. Only gender was observed as a descriptive characteristic. However, as most observed participants were students, the age of the observed participants ranged mostly between 18 and 25. At the end of the observation period, the bags in the bins were replaced by new bags in order to quantify how much waste was sorted (in)correctly, and to (re)sort the waste when necessary. After the observed waste had been quantified, the waste was disposed into the proper containers.

After a baseline assessment, a two-week intervention period started. During this intervention period, informational triangles were placed on each table at the University cafeteria. Each side of the triangle displayed pictorial information of respectively one category (paper, plastic, or general waste). All pictures showed products that were actually sold in the university cafeteria. The signs also included the information that no dirty items should go into the plastic and paper bins in accordance with the University guidelines (see Appendix A for the informational triangles employed in this study).

Lastly, after the two-week intervention period, waste separation behavior was observed again in a post-test observation, using the same procedures as during the baseline observation.

### 3.2. Results—Study 2

During the baseline observation period, 272 people were observed (66% female), separating a total of 802 items. In the post-test period, 286 people (68.5% female) were observed with 819 quantifiable items. No significant difference between baseline and post-test was found (*χ*^2^ = 2.6, *p* = 0.10). A total of 56.9% of the waste was sorted correctly at baseline (456/802) and 52.9% during the second observation (424/802). Although the intervention seemed to have increased the correct sorting for some items, the correct sorting of other items was reduced. These changes were not systematic across the different types of waste, however; no significant differences were found when analyzing the three categories separately (all *p* values > 0.05) (see Table 4).

### 3.3. Conclusions—Study 2

In a university setting, just over half of the waste was separated correctly. Given the positive intention of the students towards waste separation, there was a gap between intention and observed behavior. Comparable studies found higher correlations but used self-reports on behavior. As mentioned above, intention is often a good predictor of behavior, but not when people do not have enough knowledge or skills to perform the behavior correctly [7,14]. An intervention where extra information was made available to the students to increase knowledge and skills did not have a significant effect on improving recycling behavior. It may therefore be important to assess (1) if the intervention did indeed increase the knowledge of the participants, and (2) the reason why the waste separation behavior did not improve.

## 4. Study 3: A Post-Hoc Process Evaluation

### 4.1. Background/Methods—Study 3

To better understand the (absence of) effects from study 2, a post-hoc process evaluation was conducted. The main aim of this study was to investigate how the participants had processed the information provided in the informational triangles in Study 2, and to generate more recommendations for future recycling.

One week after Study 2 ended, university cafeteria visitors were approached on weekdays, between noon and 2.00 p.m., and were asked to fill out a short questionnaire; 59 agreed to participate. Next to demographic questions (age, gender, and position at university), participants were asked how much time they had spent in the cafeteria during the last three weeks (ranging from ‘only today’ to ‘every day’). Subsequently, participants were asked if they had noticed the informational signs (yes/no), whether they had read them (yes/no), and whether the information had changed their behavior (no/a little bit/a lot). Additionally, they were asked if their waste sorting was better/equal/worse compared to others. Lastly, participants were asked to give recommendations on how to best inform them about the correct ways of sorting waste (open-ended question).

### 4.2. Results—Study 3

From the 59 visitors, 5 were excluded because they did not spend time in the cafeteria in the past three weeks. Out of the 54 other participants (79.5% female; 20 years ±1), 8 visited the cafeteria on a daily basis (14.8%), 18 spent most days at the cafeteria (33.3%), and just over half of them (*n* = 28; 51.8%) had spent a few days in the cafeteria. Of the respondents, 48 (88.9%) had noticed the sign, and 35 (72.9% of those 48; 64.8% of the total group) actually read the information. From those who had read the sign, 80% had changed their behavior either a little (*n* = 25) or a lot (*n* = 3); 20% (*n* = 7) indicated that they had not changed their behavior at all. All participants were asked whether their recycling behavior was better, equal to, or worse than others: 23 (42.6%) answered better, 30 (55.6%) answered equal, and one person (1.8%) answered worse than others. Out of the 54 participants, 9 (16.7%) said that there was a better way of displaying information about correct sorting in the cafeteria. When asked to explain what ways they thought would work better, seven suggested that the information could be displayed on or near the bins, rather than on the cafeteria tables, while two suggested that the information should be distributed on the student portal.

### 4.3. Conclusions—Study 3

The results indicated that the signs were not consistently ready by the participants. Reading the signs helped students to change their waste-sorting behavior to some extent. However, some students suggested that more information should be available where the actual decision making takes place. Additionally, we found that people estimated their own sorting behavior to be better than, or at least as good as, that of others. This might be one explanation for why people tended to be less engaged in changing their sorting behavior.

## 5. General Discussion

We showed in three studies—a survey, an intervention, and a post-test—that students had a positive attitude towards recycling and had the intention to behave pro-environmentally. However, results also show that there was a gap between the intention of the students to sort waste correctly and their actual behavior. Based on the survey, the intervention strongly focused on the knowledge aspect of recycling, and thus the self-efficacy determinant of the RAA model. The findings show that although the information provided in the intervention was noticed by the students, it was not often read. This can be a reason for the ineffectiveness of the intervention. Incidental comments from the students with regard to the provided information suggested that new interventions could focus on triggering them to make the right waste separation decisions where the actual decision making takes place.

Comparing the results of Studies 1 and 2, it seems that while knowledge of recycling helped participants to sort waste at a near-perfect level for some items, and well above chance for others (Study 1), actual sorting behavior was a lot less accurate (Study 2) and the intervention even reduced correct sorting of some items. Thus, it seems that the information on how to sort the waste was confusing for the participants; moreover, the provided information was observed but not read. Future research could improve this by monitoring what the most frequent waste products are and giving people clear pictures and instructions, which could vary depending on the setting. A department coffee lounge is different from a restaurant in that respect. Moreover, some products are extremely confusing, such as paper-like napkins with a written message of “from 100% recycled paper”, which are not supposed to go into the paper bin.

The three studies presented in this paper complement each other in terms of insight into the role of determinants of recycling behavior. However, it also has some limitations. Firstly, the intervention focused strongly on the knowledge component, following the results of Study 1. As a result, the other determinants of intention, attitude, and subjective norm were not considered. As organizational research has shown that norms and identification play an important role in the adoption of behavior change in organizational behavior and organization-wide changes [20,21,22], future interventions may consider targeting these determinants.

Secondly, participants’ use of the knowledge provided in the intervention was unclear. In Study 1, participants indicated that they did not see harm in mis-sorting some items if they sorted the majority of items right. In Study 3, participants who indicated having read the information mostly changed their behavior only a little. This is reflected in research on moral compensation, which shows that people allow themselves immoral behavior after doing something moral [23,24]. This moral compensation effect could explain the discrepancy between the knowledge observed in Study 1 and the sorting behavior in Study 2. Future interventions could aim to motivate participants to a higher sorting accuracy.

Third, the relationship between the determinants, recycling intentions, and actual behavior is inferred across the different studies. Study 1 provided a clear insight into the determinants of intentions, Study 2 related knowledge to actual behavior, and Study 3 connected the determinants to self-reported past behavior. Future research should, however, focus on testing the full model in the context of one study. The number of participants was relatively low (121 and 59). We did not have a minimum number in advance; we asked participants through social media, and in the university cafeteria, within a fixed period (on 11 days in April 2018 for Study 1, and on 30 May 2018 between 11–14 h for Study 3). We did not have specific hypotheses to test; the study was explorative. We think that the outcomes are clear enough to be meaningful, even with these less than ideal numbers of participants.

Fourth, all our participants were university students. However, there is no reason why the outcomes of this study could not be generalized to other organizations, to departments of organizations, and to members of those organizations. Moreover, further research should make better use of observational measures of actual behavior instead of self-reports on behavior.

## 6. Conclusions

In conclusion, the implemented waste sorting system was not successful at prompting people to recycle their waste correctly. The intervention applied in this research did not improve the behavior, but yielded some useful insights. As suggested by the participating students in the qualitative part of Study 3, the most promising approach would be to provide the visitors with information on each product in terms of paper, plastic, or general waste on the product itself. One participant suggested putting stickers on the products that correspond with the colors of the bins. In that case, students would be immediately nudged into the right behavior, at the moment the decision process takes place, and the confusion that was observed in Studies 1 and 2 may be reduced as a result. However, even in that situation, clear visual information is also needed on paper and plastic waste that is to ‘messy’ to be recycled. In sum, although positive intentions typically appeared to be in place, new interventions should be developed to improve waste separation behavior.

## Figures and Tables

**Table 1 ijerph-16-00093-t001:** Questionnaire to measure social cognitive determinants including intention, barriers, and past behavior.

Determinant	Scoring
Attitude (α = 0.63)	
1. Me separating my waste accurately during lunch breaks for the next 3 months is	1—Bad; 7—Good
2. Me separating my waste accurately during lunch breaks for the next 3 months is	1—Unpleasant; 7—Pleasant
Social Norm (α = 0.18)	
3. Most people who are important to me think that I should separate my waste accurately during lunch breaks for the next 3 months	1—False; 7—True
4. Most students at Maastricht University separate their waste accurately during lunch breaks for the next 3 months	1—Disagree; 7—Agree
Self-Efficacy (α = 0.78)	
5. I am confident that I can separate my waste accurately during lunch breaks for the next 3 months	1—False; 7—True
6. Separating my waste accurately for the next 3 months is	1—Difficult; 7—Easy
Barriers 1	
7. (In case of score 1 or 2 on question 6) Why is separating your waste difficult?	Open-ended question
Intention (α = 0.81)	
8. If I really wanted to, I could separate my waste during lunch breaks for the next 3 months	1—Unlikely; 7—Likely
9. I intend to separate my waste accurately during lunch breaks for the next 3 months	1—Definitely not; 7—Definitely
10. I expect to separate my waste accurately during lunch breaks for the next 3 months	1—Unlikely; 7—Likely
Past Behavior (self-reported)	
11. In the past 3 months, how often have you separated your waste accurately during lunch breaks?	1—Never; 7—Always
Barriers 2	
12. Name at least one thing that prevents you from separating your waste accurately during lunch breaks for the next 3 months.	Open-ended question

**Table 2 ijerph-16-00093-t002:** Results on sorting waste pictures into plastic, paper, or general waste (*n* = 106). The right answer is highlighted in grey.

Product	Plastic		Paper		General Waste	
	%	*N*	%	*N*	%	*N*
Plastic						
Plastic utensils	99	105	0	0	1	1
Plastic cups	100	106	0	0	0	0
Plastic candy wrappers	77	82	0	0	23	24
Plastic coffee cup lids	99	105	0	0	1	1
Paper						
Coffee cups (paper)	2	2	58	61	40	43
Paper candy wrappers	13	14	73	77	14	15
Paper sandwich boxes	0	0	90	95	10	11
General Waste						
Noodle boxes	6	6	52	55	42	45
Food waste (*N* = 105)	0	0	0	0	100	105
Candy wrapper (mixed materials)	23	24	15	16	62	66
Chip bags	55	58	0	0	45	48
Sugar sticks (*N* = 105)	7	7	53	56	40	42
Sandwich bag (mixed materials) (*N* = 105)	52	55	0	0	48	50
Napkins (paper at the end of the recycling cycle)	1	1	82	87	17	18

Colorized: the percentage and number of correct sortings.

**Table 3 ijerph-16-00093-t003:** Determinants predicting waste separation: means and standard deviations, correlations with intention, and regression to predict intention.

Determinant	Mean (*SD*) (Range 1–7)	Correlation with Intention	b (SE)	Standardized Beta (t)	b (SE)	Standardized Beta (t)
Intention (*N* = 105)	6.30 (0.76)					
Attitude	5.78 (0.99)	0.63 *	0.27 (0.31)	0.36 (4.91) ***	0.22 (0.06)	0.29 (4.00) ***
Injunctive norm (*N* = 105)	5.13 (1.65)	0.47 *	0.14 (0.03)	0.30 (4.71) ***	0.13 (0.03)	0.28 (4.47) ***
Descriptive norm	3.78 (1.38)	0.10	-	-	-	-
Self-efficacy	5.51 (1.15)	0.63 *	0.27 (0.05)	0.40 (5.60) ***	0.25 (0.05)	0.37 (5.38) ***
Past behavior	5.7 (1.1)	0.52 *			0.15 (0.05)	0.22 (3.27) **
***F***				54.29		47.34
***R*^2^**				0.62		0.66

* *p* < 0.01; ** *p* = 0.002; *** *p* < 0.001.

**Table 4 ijerph-16-00093-t004:** (In)Correctly sorted waste at T0 and T1.

Product	T0		T1		*χ*^2^; *p*
	% Correct	*N*	% Correct	*N*	
Plastic	69.7	208	70.4	180	0.06; 0.81
Plastic bottles/cups	90.0	10	91.7	12	
Plastic utensils	50.5	105	50.0	86	
Plastic wrappers	78.3	83	73.4	64	
Plastic—other	60.0	10	66.7	18	
Paper	58.6	122	59.3	88	0.40; 0.53
Paper sandwich boxes	77.3	22	58.3	24	
Office papers	100	18	100	8	
Paper coffee cups	28.8	59	57.9	19	
Paper bags	66.7	18	72.7	11	
Paper—other	20.0	5	7.7	26	
General waste	52.3	472	50.3	534	2.00; 0.16
Dirty plastic	69.1	55	16.9	64	
Dirty paper	47.5	40	45.5	11	
Dirty cups	35.3	17	38.7	75	
Food waste	84.6	52	96.6	59	
Noodle boxes	40.0	5	20.0	5	
Plate (paper and plastic coating)	44.4	45	49.0	49	
Sauce packages/sugar sticks/salt packages	59.1	22	61.1	18	
Aluminum/tetra pack	73.9	23	78.9	19	
Napkins	44.8	125	46.8	124	
Bags (mixed materials	37.5	8	35.7	14	
Bamboo utensils	52.9	34	78.6	14	
Sugarcane plate	50.0	18	48.3	29	
Plastic with (aluminum sticker)	35.7	14	37.5	8	
General waste other	57.1	14	46.9	49	
Total	56.9	802	52.9	819	0.26; 0.61

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
