# Peer review of "Waste Separation in Cafeterias: A Study among University Students in the Netherlands"

_ijerph, 2018, doi:10.3390/ijerph16010093_

Round 1
Reviewer 1 Report
This is an interesting piece of research with wide applications across the university sector, but also across other business sectors with on-site catering (government and private sector offices, factories etc.). In my view, the paper should be published, but I would like to see the following changes made to improve its quality and impact outside the research community.
Title
The research applies to cafeteria waste rather than waste produced by students across the board. Therefore, I suggest amending the title to reflect this. Perhaps to
“Waste separation in cafeterias: a study among university students”
Line 15
Change “… the amount of greenhouse gasses…” to “the production of greenhouse gases...”
Line 17
Clarify that the research relates to cafeteria waste rather than waste produced by students in general.
Lines 28 - 30
Add a brief comment about the extent that these results apply to other organisations.
Line 36
Clarify that all municipal and related waste that is burned in the Netherlands is processed in systems that recover energy from the burning waste.
Line 63
Again, please clarify that the survey related to cafeteria waste rather than (say) waste from student residences.
Lines 127 – 135
Here (or in the general discussion) is it possible to compare these results with findings from any other similar surveys in the literature?
Overall discussion
The findings should be put into context through comparisons with the literature. Also, the question of the extent to which these results apply to non-students should be discussed. Are there any lessons to be applied in other cafeterias? Will the students’ behaviour be any different from other young people or from older populations? Etc etc.
Reviewer 2 Report
The studies carried out in large part (Stud 1, Study 3) were based on questionnaires. Therefore, the method of preparing this surveys be clearly presented:
-how the questions were developed,
-on what basis the scale of assessments used was adopted,
-how the interpretation of the obtained test results has been determined,
-whether the number of respondents is sufficient to recognize the test results as meaningful.
Round 2
Reviewer 2 Report
The Authors clearly defined the methodology and scope of the research. The work can be treated as a case study. In its current form it is suitable for publishing.